# Bridging the Gap Between Climate and Health Systems: The Value of Resilience in Facing Extreme Weather Events

**DOI:** 10.3390/ijerph22081258

**Published:** 2025-08-11

**Authors:** Eloïse Dagneau, Sophie Marie Ehgartner, Gabriel Gulis

**Affiliations:** 1Access-To-Medicines Research Center, KU Leuven University, Vlamingenstraat 83, 3000 Leuven, Belgium; 2Independent Researcher, 2000 Frederiksberg, Denmark; 3Department of Public Health, University of Southern Denmark, Degnevej 14, 6705 Esbjerg, Denmark; ggulis@health.sdu.dk

**Keywords:** resilience, extreme weather events, climate adaptation, health systems, preparedness

## Abstract

With the rise in climate emergencies come complex challenges for the field of health systems that require equally complex strategies. Acknowledgement of the pressures experienced by health systems in relation to CC is increasing, but it remains understudied. This article aims to strengthen the link between climate and health systems by examining how health systems can incorporate resilience strategies to better respond to extreme weather events. Framed by an explorative literature search, the policy analysis dives into the content of the Health National Adaptation plans of both Fiji and Tanzania in relation to the WHO Operational Framework for building a climate-resilient health system. Our findings suggest updating the WHO Framework by adding additional elements under each component and by adding new components, namely human mobility and humanitarian aid. We recommend following a step-by-step ‘system approach’ to facilitate the integration of resilience building within health systems and to acknowledge the iterative aspect of this process.

## 1. Introduction

The increasing frequency of exposure to climate-related extreme weather events (EWEs) poses a particular challenge to health systems (HSs). These emergencies have drawn attention to the complex problems they cause, especially given their often highly harmful impacts [1,2]. EWEs have been identified as one of the top three contributors to climate-related health impacts with floods and storms accounting for 68% of these health impacts [3,4]. The intricate interplay between climate and health contribute significantly to the burden of diseases through direct, indirect, and mediated pathways [5,6,7]. In particular, the negative impact on HS highlights pre-existing vulnerabilities.

HSs are not only responsible for routine care; their role goes beyond as they are also critical in responding to global public health threats, such as EWEs [8]. The confluence of heightened pressures, additional costs, and reduced access and performance hinders its ability to fulfill their crucial role [2,7,9]. Rather than simply bearing the weight of these challenges, resilience offers a way to strengthen HS—helping them absorb, adapt, and respond to disturbances more effectively [10,11].

A resilient HS, by definition, possesses the ability to maintain its functionality under stress by continuing to provide essential services before, during, and after crises. This resilience, in turn, ensures continued efficiency and responsiveness in the delivery of essential services [10,11,12].

Due to the sharp rise in climate-related emergencies, it is vital to explore ways to adapt to climate change in order to build climate-resilient HS. Currently, countries are encouraged by the World Health Organization (WHO) to follow the WHO Operational Framework for Building Climate Resilient Health Systems (WHO Operational Framework) [11]. The WHO Operational Framework was developed in response to growing calls from member states and partners for structured guidance on tackling the health impacts of climate change. It provides a roadmap for health systems to become more resilient and responsive to both immediate climate variability and long-term climate trends. The framework is grounded in 10 key components that aim to strengthen the ability of health authorities to anticipate, prevent, and manage climate-related health risks [11]. The framework supports countries in meeting global commitments like the United Nations Framework Convention on Climate Change (UNFCCC) and Sustainable Development Goals (SDG) by guiding the integration of climate resilience into national health plans—especially in low-income countries and those developing health components of National Adaptation Plans. Without prescribing a fixed timeline, it offers a flexible, phased approach that encourages cross-sector collaboration to strengthen and future-proof health systems against climate risks [11].

Since the adoption of the WHO Operational Framework, the literature has continued to contribute to the topic, offering various recommendations on how to integrate resilience. These are described in the following section.

**Leadership and Governance:** During disasters, unprepared governments struggle with unclear decision-making, often resulting in a lack of coordination and unclear responsibilities [13,14]. Leaders must use policy levers such as climate change adaptation plans (CCAPs), and ensure synergies between all sectors [4,15,16]. Transdisciplinary approaches and clear communication contribute to improved government legitimacy, faster and more confident decision-making [1,10,17].

**Healthcare Workforce:** These first responders often become increasingly vulnerable during disasters. They face challenges such as unusual working conditions, patient overload, and understaffed departments [18,19]. HCWs working under abnormal conditions cannot deliver care ‘as usual’. New stressors such as altered heat, flooded equipment, and inadequate ventilation may contribute to lower performance. These occupational hazards are a threat to the physical and mental well-being of HCW [20]. HCWs must be trained on how to cope with long-term shocks and to prioritize their own well-being [13,20]. Additionally, funding for training and sufficient human resources are needed to better organize and retain a dynamic workforce [4,20].

**Service Delivery and Essential Medical Products and Technologies—Climate Resilient and Sustainable Infrastructure:** Damaged or destroyed health infrastructure reduces the quality of care and the number of patients admitted [21,22]. Damaged transport and facilities are unable to meet increased demand, which worsens the working conditions of HCWs [9,19,20]. A disaster damages not only hospital infrastructure, but also extended health facilities [22,23]. These additional health-related sectors all contribute to the functioning of societies by providing access to clean water, nutritious foods, and energy to function [15]. The entire network of infrastructure would require reorganization and strengthening. Indeed, changing the design of urban and rural spaces is a recurring theme in the literature [9,10,14,22,24,25].

**Supply Chain Resilience Pathway—Management of Environmental Determinants:** HSs operate in larger systems that include sectors such as agriculture, water, energy, housing, and transportation [11]. HSs must collaborate with different sectors whether through joint monitoring of exposures and impacts, prevention plans, or shared policy advice [9,20,21]. This intersectoral approach would improve efficiency but requires active coordination [11].

**Health Information System (HIS):** When data collection, analysis, and feedback channels are disrupted, HISs cannot provide timely and reliable information [5]. The disruption creates a blindfold effect that results in high uncertainty in guiding responses to health emergencies [13,20]. Reinforced information on vulnerabilities, existing capacities and adaptation assessments is needed especially in the context of data integration of climate, environmental and epidemiological data [6,26,27]. Also, intersectoral early warning systems and communication platforms improve data tracking efficacy and informs the research agenda for climate change and health [11,20,26].

**Financing:** Only 2% of climate financing is allocated to health [11]. Countries need to reduce costs over the long term by building climate-smart healthcare, implementing cost-effective community programs, creating multipurpose solutions with co-benefits. Finally, money and resources are managed by those who make decisions. Having the necessary resources does not imply success nor does having insufficient or scarce resources mean failure. It is often the bureaucratic processes that are criticized in the literature as causing a major gap in financial resilience.

Against this background, this paper aims to address the gap between the content of the WHO framework’s recommendations and the literature in building national health systems resilience plans. First, the article will describe the methodology behind the literature search, the country selection for the policy analysis and the creation of the question-guide. Secondly, it will present the main strengths and challenges of the country’s policy content. Then, it will discuss these findings, highlighting areas of improvement for the WHO framework. It will conclude by targeting areas for improvement and with the need for more attention.

## 2. Methodology

This paper analyzes the content of Fiji’s and Tanzania’s national health adaptation plans to assess the extent to which they incorporate resilience strategies. The analysis builds on research conducted during the authors’ joint master’s thesis, completed in 2023 [28]. To achieve this objective, we conducted a literature review and developed a custom tool to guide the content analysis of the selected plans.

### 2.1. Exploratory Literature Research

An exploratory literature review was conducted to identify the current influences of climate change-related exposures on the resilience of health systems. The research focused on the following keywords: ‘health systems’, ‘extreme weather events’, and ‘resilience’. We searched PubMed, Medline (Ovid), Scopus, and Cochrane Library databases using all three terms. The inclusion criteria limited results to articles published between 2013 and 2023, in English, and with full-text availability. We then used a snowballing technique to include other relevant sources.

### 2.2. Country Selection

For the policy analysis, countries were included if they had published a Health National Adaptation Plan (HNAP)—or a comparable document under a different title—that aligned with the WHO Operational Framework. We reviewed all 193 United Nations Member States to identify such documents, focusing on those that addressed the three core concepts identified in the literature review. Only official English-language publications were considered, as both authors are fluent in English. The identified countries were categorized into quartiles based on their health expenditure per capita (in USD), allowing for comparisons within similar economic context [29]. To remain within a feasible scope for in-depth analysis, two countries were selected: the Republic of Fiji and the United Republic of Tanzania—both representing the lowest quartile of health spending while fulfilling all inclusion criteria.

### 2.3. Policy Document Analysis

To assess the selected policy documents, we used the WHO’s 2015 Operational Framework [11] as the analytical foundation and supplemented it with additional questions derived from the literature review. These questions were organized under the corresponding components of the WHO Framework—expanded from 10 to 11 in our analysis (Overarching Considerations, Leadership and Governance, Health Workforce, Vulnerability, Capacity, and Adaptation Assessment, Integrated Risk Monitoring and Early Warning, Health and Climate Research, Climate Resilient and Sustainable Infrastructure and Technologies, Management of Environmental Determinants of Health, Climate-Informed Health Programs, Emergency Preparedness and Management, and Climate and Health Financing)—and further supplemented by newly identified thematic categories based on our review findings (Human Mobility and Displacement and Humanitarian Aid) (see Appendix A for the full question guide).

The purpose of the question guide was not to evaluate resilience capacity as a whole, but rather to determine the degree to which climate resilience measures were explicitly included in the policy texts. Each policy document was assessed for alignment with the WHO Operational Framework and for the presence of elements highlighted in the literature review. We used a coding system to indicate whether a question was fully addressed (+), partly addressed (+−) or not addressed (−). This structured approach allowed us to evaluate the integration of climate resilience into national health planning, identify existing gaps, and propose recommendations for strengthening the WHO Framework.

Ultimately, this article aims to contribute to the understanding of how climate change affects health system resilience and to explore the adaptation strategies currently embedded in national health policy frameworks.

## 3. Results

The first step of our research involved identifying innovative insights from the literature on components missing from current resilience frameworks. A total of 66 articles were included in the review. As governments began implementing Health National Adaptation Plans (HNAPs), analysis of both the implementation process and associated publications provided valuable new perspectives [7,11]. It revealed recurring themes that we have grouped into two broader components, namely:

(1) Human mobility (missing component): Extreme weather events (EWEs) can cause loss, destruction, and instability, leading to high rates of displacement [30]. While the unpredictability of migration challenges governmental responses, integrating human mobility into HS can ensure equitable access to healthcare for all, regardless of migration status. Recognizing the rights of migrants and establishing effective financing mechanisms can prepare health systems for the pressures of displacement [20,31].

(2) Humanitarian aid (missing component): Schwerdtle et al. (2020) propose a framework for climate-resilient humanitarian health systems to help organizations integrate resilience into policies and strategies [31]. This approach enhances the capacity to manage and mitigate climate-related health risks, which is critical for timely and effective response. However, humanitarian aid systems face delays in funding and lack a global framework to address the health challenges of migration. Migrants are often excluded from care, exacerbating inequalities and criminalizing those who seek help. Addressing these systemic gaps requires a comprehensive, system-level approach to meeting health needs of displaced populations. Strengthening humanitarian response with targeted support for migrants is vital to build resilience against increasing climate-related crises [20,29,30].

Although the 44 policy-guiding questions from the WHO Operational Framework are grounded in empirical evidence, the literature reveals important gaps and nuances not captured within the original components. In response, we integrated the two newly identified themes—human mobility and humanitarian aid—and five further questions (Appendix A) into our policy analysis tool to complement and further refine the WHO framework. An overview for the updated framework is listed in Table 1.

In the second phase, we identified countries to serve as case studies. Of the 193 UN Member States reviewed, 45 had published a National Adaptation Plan (NAP), HNAP, or equivalent policy document by February 2023. We focused on English-language documents that addressed core themes from the literature—health systems, resilience, and extreme weather events. A total of 27 plans met these criteria and were categorized into quartiles based on health expenditure per capita (USD), using data from the WHO Global Health Expenditure Database [29]. After a detailed review, four documents (from two quartiles) followed the WHO Operational Framework. One was excluded due to an “approval pending” watermark, and another due to time constraints. The final analysis was conducted on two countries—the Republic of Fiji and the United Republic of Tanzania—using our extended question guide.

Country 1 (Fiji): The document titled “Republic of Fiji National Adaptation Plan: A Pathway towards Climate Resilience of Fiji” [32] addressed 31 out of 51 questions (60.78%) fully. Ten questions (19.61%) were insufficiently addressed, and ten (19.61%) were not addressed at all. An overview of the suggested elements in the HNAP of Fiji can be found in Figure 1.

Country 2 (Tanzania): The policy titled “Health—National Adaptation Plan (HNAP) to Climate Change in Tanzania 2018–2023” [33] fully addressed 34 out of 51 questions (66.67%). Five questions (9.80%) were insufficiently addressed, eleven (21.57%) were not addressed, and one (1.96%) was deemed not applicable. An overview of the suggested elements for the HNAP of Tanzania can be found in Figure 2.

Adherence to WHO Operational Framework Questions (n = 44):Fiji fully addressed 30 questions, gave insufficient responses to 9 (20%), and failed to respond to 5 (11%)—a 68% full alignment rate.Tanzania fully addressed 32 questions, with 5 insufficient (11%) and 6 unanswered (14%)—a 73% full alignment rate.

Response to Additional Literature-Based Questions (n = 7):Fiji fully addressed one question (14%), gave an insufficient response to one (14%), and did not address five (71%).Tanzania fully addressed two questions (29%) and did not respond to five (71%).

The percentage breakdown, which can be taken from Table 2, offers a nuanced understanding of each country’s approach to building health system resilience. Both Fiji and Tanzania demonstrated substantial alignment with the WHO Operational Framework, fully addressing 68% and 73% of the framework’s questions, respectively. However, the markedly lower response rates to the literature-based questions—14% for Fiji and 29% for Tanzania—indicate that emerging areas such as human mobility and humanitarian aid remain insufficiently addressed in current national strategies. These findings underscore the importance of incorporating both established frameworks and evolving evidence into assessments of national climate-health adaptation efforts.

## 4. Discussion

This paper complements the WHO Operational Framework by identifying underdeveloped or missing elements that limit its practical applicability. While the framework offers a strong foundation, it lacks operational depth in several key areas, including the integration of overlapping policy domains (DRM, CCA, SDGs) within Leadership and Governance, the emotional and structural support of healthcare workers under Health Workforce, the differentiated needs of urban and rural settings within Climate-Resilient Infrastructure, enforcement mechanisms and communication strategies under Environmental Determinants, and financial autonomy and donor coordination in Climate Health Financing. Additionally, two critical dimensions—Human Mobility and Displacement and the role of Humanitarian Aid—are entirely absent, despite their growing relevance. Our proposed additions aim to fill these gaps and make the framework more context-sensitive, actionable, and responsive to real-world challenges. The following section discusses the above-mentioned complementary points.

### 4.1. Leadership and Governance

There is no one-size-fits-all model for leadership, but traits like effective resource management, strategic alignment, and transparency are essential for strengthening (HS) resilience. Success depends on linking key agendas such as disaster risk management (DRM), climate change adaptation (CCA), greenhouse gas (GHG) mitigation, and the Sustainable Development Goals (SDGs), which fosters more integrated and structured policy-making.

While improved coordination across these areas can help reduce inefficiencies [34], it is important to recognize that not all redundancies are problematic. Excessive fragmentation may waste resources [1], but strategic redundancies—such as overlapping responsibilities or systems—could actually strengthen resilience by providing backup capacity and flexibility in times of crisis. Rather than eliminating all overlap, leadership could distinguish between harmful duplication and beneficial redundancy.

Effective intersectoral communication and coordination remain vital but should be pursued with a clear understanding of the trade-offs between efficiency and adaptability. Fiji’s 2018 National Adaptation Plan (NAP), for instance, exemplifies this approach by aligning with key SDGs like clean water (SDG 6) and sustainable tourism (SDG 8) to promote both environmental and health resilience [32].

Finally, stronger accountability mechanisms—such as those in Tanzania’s H-NAP—enhance transparency and financial governance, helping to build public trust and ensure systems can respond effectively under pressure [33].

### 4.2. Health Workforce

During crises, health systems often operate under immense strain, leaving little room for flexibility. While it is easy to suggest idealistic solutions, past emergencies—such as the COVID-19 pandemic—have made it clear that meaningful progress in resilience must begin by prioritizing healthcare workers (HCWs) [35]. Supporting HCWs involves more than just education and training; it requires addressing external factors that affect their ability to perform. Mental health support, safe working conditions, and recognition of their contributions remain underappreciated but essential components of resilience. Providing psychological services helps HCWs manage stress, while adequate staffing, proper rest, and reliable access to water, electricity, and waste management promote well-being and job satisfaction. Yet, only 26 of 195 countries have developed official guidelines for occupational safety [19]. Expressing gratitude and offering fair compensation can further boost morale and strengthen long-term commitment. Indeed, the range of possible improvements concerning healthcare workers can feel overwhelming. More research is needed to identify the most urgent and accessible opportunities—the low-hanging fruit—for strengthening healthcare worker resilience.

### 4.3. Climate-Resilient and Sustainable Infrastructure

There is currently high interest in financing CCA and transitions towards green healthcare [19]. The advantage of this global effort is that solutions are provided from diverse settings experiencing a wide range of emergencies. There is a possibility to leverage international collaboration and knowledge transfer to develop a network of solutions. However, the issue of generalizability would need to be considered. Creating hubs for regional advice rather than national or global advice could provide more specific and relevant support.

Part of this new research covers the impact of EWE on construction and the built environment in cities. The indicators of damage vary according to the type of damage: *direct and tangible* (i.e., destruction of crops), *direct and intangible* (i.e., cultural heritage), *indirect and tangible* (i.e., direction of public services) and *indirect and intangible* (i.e., loss of trust) [36]. Bignami’s article reports on the need to involve the field of urban planning and design in order to reduce damages caused by disasters and to create complementary synergies with the disaster risk reduction strategy [36]. According to the UNDP, unplanned growth increases risks [37]. As the population grows and more people are living in cities, recognizing the vulnerabilities and managing the complex impacts of this demographic shift will enable greater resilience.

On the other hand, agricultural infrastructure is most affected in rural areas, as opposed to urban construction in cities. Agriculture is reported to be the main human activity affected by climate change, especially in poor rural environments. Capacity building in this sector relates to improving water supply infrastructure, improving roads through better drainage systems or road elevation, protecting river banks by adopting less harmful agricultural practices, and developing technologies or tools that can help manage insufficient water or rainfall [38,39]. There is also focus put on addressing transportation and its relationship with community isolation in rural areas [40]. Overall, infrastructure must be innovative and maintained in both urban and rural settings.

Technologies can complement the suggested solution. Promoting sustainable technologies within the EWS can be beneficial by promoting timely response, mitigating damage, and making the system better prepared [6,26,27]. For example, technological tools such as AI and GIS can be used to collect more data and intersect these to inform urban planning. Furthermore, the current “twin transition” would require finding ways to develop technologies that support sustainability. Once created, these technologies should be monitored and evaluated on all fronts [41]. However, careful planning is required to ensure that training is well developed alongside technological development. Community training on the use of risk mapping and vulnerability assessment technology has been performed before and has shown great potential. Technology brings change that may not be accepted, and this requires special attention to ensure full adoption [42].

All these struggles underline the interdependencies between sectors: water, energy, infrastructure, transport, health and well-being, etc. Not only does it underline the need to adopt a system perspective to identify the links but also highlights the need for strong leadership and governance, as mentioned above, to coordinate and build synergies. Furthermore, exploring the area of enabling environments by using local strengths and reducing costs should be considered, but more research is needed to integrate this concept into resilience building [43].

### 4.4. Management of Environmental Determinants

Collaboration in the management of environmental determinants of health is thoroughly acknowledged in both HNAPs. However, coordination and enforcement of this integrated approach remain challenging [33,44]. While sectors are generally aware of the need to collaborate, this intention is often not fully realized in practice. This gap can be attributed not only to limited communication but also to more entrenched structural and institutional barriers—such as siloed governance systems, competing mandates, and resource constraints—which hinder meaningful cross-sectoral action. Therefore, while improved communication strategies may enhance awareness and foundational understanding of the benefits of collaboration [26], they must be complemented by efforts that address these deeper systemic obstacles.

Dissemination strategies tailored to departments that struggle with implementing collaboration can play an important role—but they must be grounded in a realistic understanding of what collaboration entails. Effective integration across sectors demands time, coordination, and political will, and it often faces significant structural and institutional barriers. As such, collaboration should not be viewed as an easy or universally applicable solution, but rather as a complex, resource-intensive process that may produce uneven or delayed outcomes.

Even so, mechanisms like health impact assessments (HIAs) remain essential. HIAs not only help reinforce the importance of incorporating health considerations into all policies but also serve to identify the most practical and strategic opportunities for intersectoral coordination [33,44]. Communicating these priorities clearly can support policymakers in fostering more accountable and adaptive governance. In parallel, ongoing monitoring of policy coherence and collaboration efforts allows for reflection on past strategies, learning from implementation gaps, and adapting approaches to better fit the realities of specific contexts.

### 4.5. Climate Health Financing

The first step in achieving financial resilience is to assess available resources and finances to then identify gaps [11]. Both countries have estimated budgets in their respective HNAPs, which is helpful in identifying needs. However, they do not specify a management team, which could lead to coordination struggles and potential unsuccessful management of funds. Tanzania is aware of this as they report not having funds to afford a management team, which they recognize as a gap [33,44].

When national resources are insufficient, this can lead to an over-reliance on donations. While donors are always welcome and needed in emergencies, relying solely on international aid is not resilient [6]. Funding should make sure to promote independence and sustainable solutions. To promote true autonomy, humanitarian aid and donations should not only provide resources for primary care but also fund the training of local HCWs [45]. Cost-effective research is needed to inform donor agencies about the ranking of different interventions that they could help finance. Additionally, donor agencies often impose strict funding criteria, which make it difficult to access funds. Instead, donors should make their funding more flexible. This would in turn encourage more community involvement [16,20].

Finally, the values behind financial management can influence financial outcomes. Trust and transparency are important not only for encouraging patients to use health services, but also for good governance [16,20]. Transparent budgeting allows citizens to better understand the funding mechanisms behind interventions thereby fostering trust towards decision makers. Cascading effects include increased community involvement. When people see that their government is performing at its best, they may want to perform at their best in return. Transparency can also encourage more donations because donors can trust that their donations will be used appropriately. Overall, we argue that transparent, and flexible funding can better promote a country’s independence.

### 4.6. Missing Components of the WHO Operational Framework

This section discusses components identified from the literature as being necessary to add to improve the outdated WHO Operational Framework.

#### 4.6.1. Human Mobility and Displacement

As emergencies and EWEs become more frequent and severe, they increasingly drive human mobility and displacement, which in turn raise critical public health concerns [46]. Health National Adaptation Plans (HNAPs) must acknowledge that such crises often result in population movement and should incorporate strategies that reflect the interconnected nature of our globalized world. Human mobility presents specific challenges for disease control: increased travel—particularly via air—facilitates both the more frequent emergence and the wider spread of infectious diseases [47].

It is therefore essential to understand not only the patterns of mobility (why, where, and under what conditions people move), but also the characteristics of mobile populations (such as socioeconomic status, occupation, and education) [48]. Climate change intensifies disease transmission risks by both increasing the health consequences of displacement and by amplifying the drivers of mobility.

While emergency preparedness is a distinct and well-established component of the WHO Operational Framework, it is important to also consider the preparedness and response capacities of host countries. Health systems in receiving regions may face increased pressure, including higher patient loads, a broader spectrum of diseases, and the need to accommodate patients with diverse languages, cultural backgrounds, and unknown medical histories [48,49]. These challenges also highlight the need for thoughtful approaches to border control and disease surveillance.

#### 4.6.2. Humanitarian Aid

There is no mention of humanitarian aid in either of the HNAPs or in the WHO Operational Framework. This is concerning as HSs often do not operate alone in emergencies. However, as humanitarian aid has the potential for high-impact interventions, it must be included in the development of a national strategy. As noted by Richardson, humanitarian aid organizations can reinforce power imbalances and negatively impact national HS in the long run [50]. By following their own agenda, using their own methods, and neglecting contextual factors, they create what can be called a “parallel system” to the existing HS. An example from the literature on pandemic preparedness and response underlines that humanitarian aid may be funded by donors who prioritize specific disease programs that may not reflect the local needs, or ignore upstream factors, such as politics, that make these targeted interventions too narrow and inappropriate to address the country’s burden of disease [45]. Additionally, these funding mechanisms are often offered on a short time period. While the short-term results may be positive, they rarely result in long-term sustainable change. In fact, international organizations have previously been asked to leave as they did more harm than good [45]. Nonetheless, there is great potential for humanitarian organizations to adopt an alternative and more sustainable approach. By trusting the capacity of the local HS and building on top of it, rather than alongside it, all stakeholders involved can better cope with disasters. Ultimately, this collaboration should promote true autonomy and build resilience [45].

#### 4.6.3. Limitations

This article holds some limitations that may have led to biases to be considered. The article focused on analyzing a single policy document per country. However, gathering all relevant policy documents can ensure a more comprehensive understanding. When policy documents are examined collectively, missing elements from the NAP analysis can potentially be found in other policy documents. Our search was also limited to the English language, which limited the author’s ability to read certain HNAPS (which, for many, were only written in the country’s national language), but also limited the scope of the explorative literature search, thereby often excluding more local or indigenous research as well as reducing the diversity of perspectives on the topic. We were also unable to complete the analysis the three countries that met all criteria due to the limited time-frame available and limited their analysis to Tanzania and Fiji.

## 5. Conclusions

The use of the WHO Operational Framework as a tool is proving to be very useful for countries wishing to develop policies such as HNAPs.

This article highlights gaps in the WHO Framework and identifies weaknesses in specific components that would benefit from further strengthening. Indeed, the HNAPS would need to address the new questions and components identified in this article to further strengthen their policies towards building resilience in health systems exposed to climate-related emergencies. Indeed, human mobility, humanitarian aid, safe and healthy working conditions for HCWs and financing strategies alongside reinforced governance are components that require more focus when developing policies.

Despite apparent gaps, Tanzania’s and Fiji’s HNAPs have made large progress in setting clear objectives for each component of their HS showcasing their system approach.

We suggest an update of the WHO Operational Framework in the sense of adding the following components: (a) human mobility and displacement and (b) humanitarian aid. The WHO Operational Framework should also improve certain aspects, particularly with regard to the following: (a) leadership and governance, (b) financing, and (c) the health workforce as these appeared to be the weaker components to be addressed in policy development. The next step for this article would be to test the implementation of the HNAP and iteratively adjust the policy.

## Figures and Tables

**Figure 1 ijerph-22-01258-f001:**
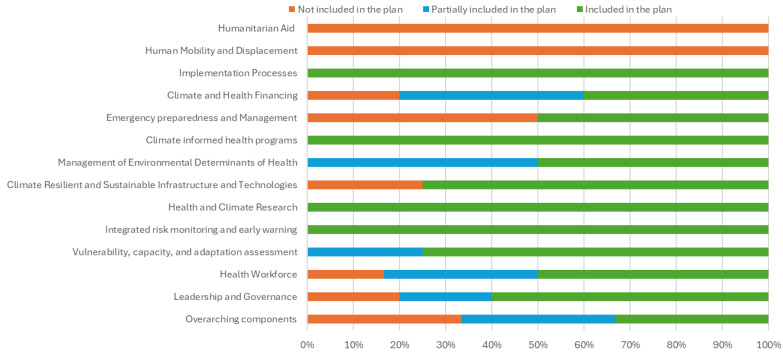
Elements of health system resilience included in the Health National Adaptation Plan of Fiji.

**Figure 2 ijerph-22-01258-f002:**
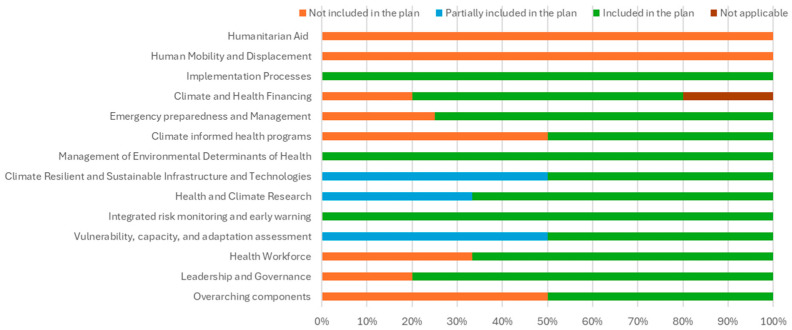
Elements of health system resilience included in the Health National Adaptation Plan of Tanzania.

**Table 1 ijerph-22-01258-t001:** Component results from the exploratory literature search to specify and update the question-guide for policy analysis of HNAPs. The added questions and new components (grey background) specify and update the question-guide from the WHO-framework to a create our updated version of the question-guide.

	All Components for an Updated WHO-Framework	Questions from the WHO-Framework	Added Questions for the Guide
WHO-Framework Components	Overarching components	5	1
	1. Component of Leadership and Governance	3	2
	2. Component of Health Workforce	5	1
	3. Component of Vulnerability, capacity, and adaptation assessment	4	0
	4. Component of Integrated risk monitoring and early warning	3	0
	5. Component of Health and Climate Research	3	0
	6. Component of Climate Resilient and Sustainable Infrastructure and Technologies	4	0
	7. Component of Management of Environmental Determinants of Health	4	0
	8. Component of Climate informed health programs	4	0
	9. Component of Emergency preparedness and Management	4	0
	10. Component of Climate and Health Financing	5	0
New	Component of Implementation Processes	0	1
	Component of Human Mobility and Displacement	0	1
	Component of Humanitarian Aid	0	1
	Total	44	7

**Table 2 ijerph-22-01258-t002:** The table summarizes the evaluation of 51 questions across two countries, Fiji and Tanzania, categorized into responses (+, +−, −) and “not applicable”. It also distinguishes between questions based on the WHO Operational Framework (44 questions) and additional literature (7 questions), presenting counts and percentages for each response type. This provides insights into each country’s engagement with health system resilience criteria.

	Total	Country 1 (Fiji)			Country 2 (Tanzania)			
		+	+−	−	+	+−	−	N.A
Questions answered (n (% of total))	51	31 (61%)	10 (20%)	10 (20%)	34 (67%)	5 (10%)	11 (22%)	1 (2%)
WHO questions (% of total)	44 (86%)	30 (68%)	9 (20%)	5 (11%)	32 (73%)	5 (11%)	6 (14%)	1 (2%)
Questions based on additional literature sources (n (% of total questions))	7 (14%)	1 (14%)	1 (14%)	5 (71%)	2 (29%)	0	5 (71%)	-

## Data Availability

The data can be freely publicly accessed on the governmental websites and on the WHO website. The data used are also available on freely accessible platforms. Global health expenditure data were sourced from the World Bank at https://data.worldbank.org/indicator/SH.XPD.CHEX.GD.ZS (accessed on 11 May 2025). All national documentation was selected from the national websites directly. These can be provided upon request from the corresponding authors.

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
