# Peer review of "Bridging the Gap Between Climate and Health Systems: The Value of Resilience in Facing Extreme Weather Events"

_ijerph, 2025, doi:10.3390/ijerph22081258_

Round 1
Reviewer 1 Report
Comments and Suggestions for Authors
The paper addresses an important topic: the resilience of health systems in the face of extreme weather events. It takes the WHO Operational Framework for Building Climate Resilient Health Systems as its point of departure, first expanding the elements covered in the framework as the result of a literature review (7 new questions, and 3 new dimensions of consideration) and then applying these to an analysis of National Health-Adaptation Plans for Fiji and Tanzania. The work appears to have been done in fulfilment of a Masters degree for the lead author. The paper will be of interest to anyone concerned with the climate resilience of health systems worldwide. The added aspects and dimensions are sound, needed, well argued for, and will be much appreciated by those undertaking the work of building climate resilience into existing health systems.
The paper would in my view be stronger if it took at bit more time in the introduction to consider the relative merits of alternative framings of resilience. As it is, a definition is proposed that is about capacity to maintain functionality under stress. Implied is a privileging of the capacity of systems to "bounce back" to 'normal' under adversity, without acknowledging that 'normal' is increasingly dysfunctional, and that what is also called for is the capacity (within health systems and society-wide) to 'bounce forward' into radically new ways of seeing, being and doing which might enable a stepping away from contemporary versions of modernity predicated on endless growth, exploitation, domination, wealth accumulation/concentration, and ecological degradation (what some refer to as the Death Economy).
For those with less prior knowledge of the WHO Operational Framework, it would be helpful if the authors said more about its genesis, roll-out, implementation, timelines, etc.
The authors sing a familiar tune about the presumed dangers of "redundancies", calling for closer collaboration. The costs of closer collaboration (which is time-consuming and doesn't always translate into actual changes in practice on the ground), as well as the very real and pervasive barriers to collaboration (given siloed structures and budgets, turf struggles, etc) are not acknowledged. Further, there are certain kinds of "redundancies" (like modularity) that are actively promoted in the resilience literature, accompanied by well-argued critiques regarding the negative impacts of undue emphasis on "efficiency". The authors would do well to engage a bit more with these critiques so as to offer a more nuanced and critical assessment.
It is to be noted that some aspects of resilience-related risk management are quite risk-specific, but also that not all eventualities can be planned for, such that 'general resilience' (Carpenter et al, 2012) might also be called for.
While many "shoulds" are offered, implying that a level of capacity (in financial and good governance terms, for example) is presumed that might not be forthcoming. Further, many health systems are already stretched, and asking health care workers to do more with less in times of adversity may not be tenable. We saw this during the pandemic; and indeed more could have been said here about what was learned about health system resilience during the pandemic.
Notwithstanding these concerns and suggestions, I find the paper to be of value and anticipate its positive reception amongst those engaged in building climate resilience into health systems.
Author Response
Comment 1: The paper addresses an important topic: the resilience of health systems in the face of extreme weather events. It takes the WHO Operational Framework for Building Climate Resilient Health Systems as its point of departure, first expanding the elements covered in the framework as the result of a literature review (7 new questions, and 3 new dimensions of consideration) and then applying these to an analysis of National Health-Adaptation Plans for Fiji and Tanzania. The work appears to have been done in fulfilment of a Masters degree for the lead author. The paper will be of interest to anyone concerned with the climate resilience of health systems worldwide. The added aspects and dimensions are sound, needed, well argued for, and will be much appreciated by those undertaking the work of building climate resilience into existing health systems.
Response 1: Thank you!
Comment 2: The paper would in my view be stronger if it took at bit more time in the introduction to consider the relative merits of alternative framings of resilience. As it is, a definition is proposed that is about capacity to maintain functionality under stress. Implied is a privileging of the capacity of systems to "bounce back" to 'normal' under adversity, without acknowledging that 'normal' is increasingly dysfunctional, and that what is also called for is the capacity (within health systems and society-wide) to 'bounce forward' into radically new ways of seeing, being and doing which might enable a stepping away from contemporary versions of modernity predicated on endless growth, exploitation, domination, wealth accumulation/concentration, and ecological degradation (what some refer to as the Death Economy).
Response 2: This is a very challenging comment, thank you! We get your point with “normal being increasingly dysfunctional” and consider it from many aspects correct, unfortunately. However, we believe that such discussion would make the manuscript more philosophical as empirical and guide it away from original objective. This subject would deserve a serious discussion, which could be for example initiated by journal editors requesting editorials on topic.
Comment 3: For those with less prior knowledge of the WHO Operational Framework, it would be helpful if the authors said more about its genesis, roll-out, implementation, timelines, etc.
Response 3: Thank you, we added a brief genesis into manuscript. You can find the changes on page 2, Introduction from line 52-65.
Comment 4: The authors sing a familiar tune about the presumed dangers of "redundancies", calling for closer collaboration. The costs of closer collaboration (which is time-consuming and doesn't always translate into actual changes in practice on the ground), as well as the very real and pervasive barriers to collaboration (given siloed structures and budgets, turf struggles, etc) are not acknowledged. Further, there are certain kinds of "redundancies" (like modularity) that are actively promoted in the resilience literature, accompanied by well-argued critiques regarding the negative impacts of undue emphasis on "efficiency". The authors would do well to engage a bit more with these critiques so as to offer a more nuanced and critical assessment.
Response 4: Thank you for this insightful comment. We agree that the topic of redundancy and collaboration requires a more nuanced discussion.
In response, we have made several changes to address these concerns:
- Acknowledging barriers: We now explicitly discuss real-world obstacles to collaboration, such as siloed institutional structures, competing mandates, and limited resources. This provides a more realistic view of the challenges involved.
- Qualified our language: We have moderated our framing of collaboration, emphasizing that while it remains a valuable strategy, it is not a straightforward or universally effective solution.
- Clarified redundancy: We revised our discussion of redundancy to distinguish between inefficient duplication and beneficial forms of modularity and backup capacity. This change aligns our argument more closely with resilience thinking and avoids an overly efficiency-driven perspective.
- Improved clarity and flow: We reduced the length and complexity of relevant sections to enhance readability and tighten the overall argument.
Please find the changes in: 4.1 Leadership and Governance, page 8-9, lines 302-323 and 4.4 Management of environmental determinants, page 10-11, Lines 402-427
Comment 5: It is to be noted that some aspects of resilience-related risk management are quite risk-specific, but also that not all eventualities can be planned for, such that 'general resilience' (Carpenter et al, 2012) might also be called for.
Response 5: Thank you for the thoughtful comment—highlighting general resilience is highly relevant given the unpredictability of extreme events. While Carpenter et al. (2012) offers valuable conceptual insights, our focus is specifically on health system resilience as broadly defined by the WHO Operational Framework. As such, we did not include this reference because it does not directly address the operational or sector-specific challenges of climate-resilient health systems.
Comment 6: While many "shoulds" are offered, implying that a level of capacity (in financial and good governance terms, for example) is presumed that might not be forthcoming. Further, many health systems are already stretched, and asking health care workers to do more with less in times of adversity may not be tenable. We saw this during the pandemic; and indeed more could have been said here about what was learned about health system resilience during the pandemic.
Response 6: Thank you for this comment, again, we fully agree. We modified the discussion on this subject in revised manuscript and sited a fitting paper. Page 9, Lines 332-337 and 343-349.
Reviewer 2 Report
Comments and Suggestions for Authors
Thank you for your valuable contribution. I appreciate your efforts in proposing a new set of components to expand the WHO operational framework. The intention to integrate the health impacts of climate-related migration is timely and necessary.
From my perspective, as someone working in air quality, climate change, and disaster risk management—I find your proposed framework promising. However, I would like to offer the following suggestions to help strengthen your manuscript and improve its practical applicability:
Ground the framework in real-world lessons.
While the three additional components (human mobility, humanitarian aid, and implementation process) are well justified conceptually, their value would increase significantly if they were backed by actual country experiences. For example, case studies from countries that have already faced migration-related health challenges could provide deeper insight and practical relevance.
Consider the readiness of host countries.
When discussing human mobility, it’s important to explore not only the movement itself but also the capacity of receiving countries. How will their healthcare systems cope? What local mechanisms exist? This dimension—preparedness and capacity at the destination—is crucial but currently underdeveloped in the paper.
Reflect on the role of external actors.
Humanitarian responses often involve NGOs and international partners. However, if such efforts are not aligned with local structures, they can disrupt existing systems. I suggest analyzing how external interventions could be integrated with local governance and what frameworks might help ensure coordination and respect for community rights.
Clarify how your proposed components interact with the existing WHO framework.
While your additions are innovative, a more direct comparison with the current WHO operational framework would help readers understand exactly what gaps you are addressing and how your model complements or challenges the existing one.
Language and flow.
The paper is generally easy to follow, but some parts—particularly in the methods and results—could be refined for clarity. I will not comment on language usage in detail since I am not a native speaker, but I encourage the authors to review the text for readability.
Overall, your manuscript addresses an urgent issue and opens the door for further development of adaptive health systems in the context of climate change. With some enhancements based on practical insights and local implementation considerations, it could become a very strong contribution to the field.
Thank you again for your work, and I hope these comments support your next steps in revising the manuscript.
Comments on the Quality of English LanguageAs a non-native English speaker, I do not provide detailed corrections regarding grammar or language mechanics. However, based on readability and clarity from a reader’s perspective, some parts of the manuscript, particularly in the methods and results sections, could be improved for better flow and comprehension. Clearer phrasing would help convey the research message more effectively.
Author Response
Comment 1: Thank you for your valuable contribution. I appreciate your efforts in proposing a new set of components to expand the WHO operational framework. The intention to integrate the health impacts of climate-related migration is timely and necessary.
Response 1: Thank you!
From my perspective, as someone working in air quality, climate change, and disaster risk management—I find your proposed framework promising. However, I would like to offer the following suggestions to help strengthen your manuscript and improve its practical applicability:
Comment 2: Ground the framework in real-world lessons.
While the three additional components (human mobility, humanitarian aid, and implementation process) are well justified conceptually, their value would increase significantly if they were backed by actual country experiences. For example, case studies from countries that have already faced migration-related health challenges could provide deeper insight and practical relevance.
Response 2: We added a reference to a set of case study examples published by International Federation of Red Cross and Red crescent societies (https://www.rcrc-resilience-southeastasia.org/wp-content/uploads/2017/11/2017-Migrant-Health-Case-Studies-AP-LR.pdf) Due to space limitation we prefer not to summarize the case studies, just provide a reference.
The reference was added into 4.6.1 Human mobility and displacement, after the first sentence of the paragraph.
Comment 3: Consider the readiness of host countries.
When discussing human mobility, it’s important to explore not only the movement itself but also the capacity of receiving countries. How will their healthcare systems cope? What local mechanisms exist? This dimension—preparedness and capacity at the destination—is crucial but currently underdeveloped in the paper.
Response 3: Thank you for this insightful comment. We agree that the preparedness and capacity of host countries are crucial aspects when addressing human mobility in the context of climate change. In response, we have revised the relevant paragraph to explicitly address this point. We also highlight the importance of integrating these considerations into emergency preparedness planning.
Was added on page 12 under 4.6.1. Human mobility and displacement, lines 468-475.
Comment 4: Reflect on the role of external actors.
Humanitarian responses often involve NGOs and international partners. However, if such efforts are not aligned with local structures, they can disrupt existing systems. I suggest analyzing how external interventions could be integrated with local governance and what frameworks might help ensure coordination and respect for community rights.
Response 4: “Ensure community approaches and voices to strengthen health action” is a key theme and component of the WHO Operational Framework. While we also recognize the critical importance of community involvement and knowledge—an aspect we emphasized in our master’s thesis—we did not find that this particular area required revision in the Framework. Therefore, it was not included in the final version of our paper.
Due to space limitations, many important points were left out—not because they lack significance, but because they are already well addressed in the Operational Framework and were sufficiently considered in the HNAPs we analyzed.
Also, we believe the above-mentioned reference includes such examples (https://www.rcrc-resilience-southeastasia.org/wp-content/uploads/2017/11/2017-Migrant-Health-Case-Studies-AP-LR.pdf)
Comment 5: Clarify how your proposed components interact with the existing WHO framework.
While your additions are innovative, a more direct comparison with the current WHO operational framework would help readers understand exactly what gaps you are addressing and how your model complements or challenges the existing one.
Response 5: We added couple of words on this in Discussion part of the manuscript: page 8, Lines 287-300.
Comment 6: Language and flow.
The paper is generally easy to follow, but some parts—particularly in the methods and results—could be refined for clarity. I will not comment on language usage in detail since I am not a native speaker, but I encourage the authors to review the text for readability.
Response 6: One of co-authors is a native English speaker and she has checked the manuscript. In terms of flow, style and clarity we revised the two critical parts (Methods and Results).
Comment 7: Overall, your manuscript addresses an urgent issue and opens the door for further development of adaptive health systems in the context of climate change. With some enhancements based on practical insights and local implementation considerations, it could become a very strong contribution to the field.
Thank you again for your work, and I hope these comments support your next steps in revising the manuscript.
Response 7: Thank you!